# Ferroptosis-Related Genes as Molecular Markers in Bovine Mammary Epithelial Cells Challenged with *Staphylococcus aureus*

**DOI:** 10.3390/ijms26062506

**Published:** 2025-03-11

**Authors:** Yue Xing, Siyuan Mi, Gerile Dari, Zihan Zhang, Siqian Chen, Ying Yu

**Affiliations:** State Key Laboratory of Animal Biotech Breeding, National Engineering Laboratory for Animal Breeding, Breeding and Reproduction of Ministry of Agriculture and Rural Affairs, College of Animal Science and Technology, China Agricultural University, Beijing 100193, China; s20223040700@cau.edu.cn (Y.X.); caumsy@163.com (S.M.); 17801115315@163.com (G.D.); sy20243040889@cau.edu.cn (Z.Z.)

**Keywords:** *Staphylococcus aureus*, mastitis, stable marker, ferroptosis

## Abstract

*Staphylococcus aureus*-induced mastitis is a significant cause of economic losses in the dairy industry, yet its molecular mechanisms remain poorly defined. Although ferroptosis, a regulated cell death process, is associated with inflammatory diseases, its role in bovine mastitis is unknown. In this study, 11 *S. aureus* strains were isolated from milk samples obtained from cows with clinical or subclinical mastitis. Transcriptome analysis of Mac-T cells challenged with isolated *S. aureus* identified differentially expressed genes (DEGs). Enrichment analysis revealed significant associations between DEG clusters and traits related to bovine mastitis. KEGG pathway enrichment revealed ferroptosis, Toll-like receptor, and TNF signaling as significantly enriched pathways. Weighted gene co-expression network analysis (WGCNA) further prioritized ferroptosis-related genes (*HMOX1*, *SLC11A2*, *STEAP3*, *SAT1*, and *VDAC2*) involved in iron metabolism. Notably, the expression levels of *HMOX1* and *SAT1* were significantly increased in *S. aureus*-challenged Mac-T cells, and this upregulation was consistent with trends observed in transcriptome data from mother–daughter pairs of cows with subclinical mastitis caused by *S. aureus* infection. Furthermore, Ferrostatin-1 treatment significantly reduced the expression of *HMOX1* and *SAT1* in *S. aureus*-challenged cells, confirming the involvement of ferroptosis in this process. This study reveals that ferroptosis plays a key role in *S. aureus*-induced mastitis and highlights its potential as a target for molecular breeding strategies aimed at improving bovine mastitis resistance.

## 1. Introduction

Based on a recent study, the global prevalence of subclinical mastitis (SCM) and clinica2l mastitis (CM) was reported to be 42% and 15% in dairy cattle, respectively [1]. Subclinical mastitis, in particular, leads to chronic reductions in milk yield and increased somatic cell counts (SCCs), directly impacting dairy farm profitability [2,3]. While genomic selection (GS) has improved breeding for mastitis resistance by leveraging genome-wide markers [4,5,6], the biological interpretation of these markers remains limited. A deeper understanding of the molecular mechanisms underlying mastitis pathogenesis could be achieved by prioritizing functional candidates for GS optimization.

Ferroptosis is an iron-dependent form of regulated cell death distinct from apoptosis, characterized by iron accumulation, lipid peroxidation, and reactive oxygen species (ROS) buildup. This process is triggered by restricted cysteine uptake and reduced glutathione synthesis, leading to cellular damage and death [7,8,9]. Recent studies have shown that pathogens like Mycobacterium tuberculosis and Salmonella enterica exploit ferroptosis to promote virulence, facilitating immune evasion and tissue invasion [10,11,12]. Ferroptosis is involved in the inflammatory response during microbial infections and may act as a host defense mechanism [13,14,15]. Preliminary evidence suggests that *S. aureus* may manipulate ferroptosis pathways to promote survival and chronic infection in the mammary gland [16,17]. The key regulators of ferroptosis, such as *SAT1* (a polyamine oxidase involved in ROS generation) [18,19] and *HMOX1* (a heme-degrading enzyme that releases free iron) [20,21], play roles in modulating inflammatory responses. Despite their known functions, the specific roles of these regulators in *S. aureus*–host interactions, particularly in mastitis, remain uncharacterized.

Studies have shown that distinct viral strains elicit strain-specific immune responses in honeybees [22] and that different gut microbiota strains in social bees can influence host health [23]. Similarly, *S. aureus* strains isolated from cows with different SCC levels may activate similar host pathways, indicating the presence of conserved genetic factors that influence mastitis progression. Building on this, this study will explore the common host genes involved in ferroptosis activation across genetically diverse *S. aureus* strains, aiming to identify stable genetic markers.

This study hypothesizes that ferroptosis is a conserved mechanism by which *S. aureus* disrupts mammary epithelial integrity, independent of strain origin. Key ferroptosis regulators, such as *SAT1* and *HMOX1*, are proposed as stable molecular markers of mastitis susceptibility, regardless of strain variation. To test this hypothesis, transcriptional responses of Mac-T cells will be analyzed in response to a panel of *S. aureus* strains isolated from milk samples of cows with clinical mastitis and varying SCC levels (high and low SCC). The objective is to identify common host genes involved in ferroptosis activation across genetically diverse *S. aureus* strains and evaluate these ferroptosis-related genes as stable genetic markers for selective breeding strategies aimed at enhancing mastitis resistance in dairy cattle.

## 2. Results

### 2.1. WGS and Virulence Factor Profiling of S. aureus from Cows with Different SCC Levels and Mastitis

Whole-genome sequencing (WGS) was performed on 11 *S. aureus* strains from the milk samples. The sequencing data quality, assessed using N50 and contig statistics, was suitable for downstream analyses (Table 1). Multilocus sequence typing (MLST) classified the strains into three known sequence types (STs): ST1 (three strains), ST97 (three strains), and ST398 (one strain). Additionally, four novel sequence types not previously recorded in the MLST database were identified. These novel types were distributed across all groups without a clear association with SCC levels or clinical mastitis cases (Table 1).

The virulence factor analysis revealed both shared and group-specific patterns among the strains (Table 2). All strains carried the cap8 family genes (*cap8e*, *cap8g*, *cap8l*, *cap8o*, and *cap8p*) and the *α-toxin* gene, which encode key virulence factors. The *SdrE* gene, associated with bacterial adhesion, was detected in seven strains, primarily in the low-SCC groups (L1-L4) and the high-SCC groups (H1-H4), but was less common in the clinical mastitis group (M1-M3). In contrast, the *vWbp* gene, encoding von Willebrand factor-binding protein, was detected in strains from the high-SCC and clinical mastitis groups but was absent in strains from the low-SCC group. Strain M1 from the clinical mastitis group exhibited the fewest virulence factors, with only *cap8* and *α-toxin* genes detected. There was no consistent correlation observed between the MLST sequence types, virulence factor profiles, or SCC-based groupings.

### 2.2. Transcriptional Analysis of Mac-T Cells Challenged with S. aureus from SCC and Mastitis Groups

To evaluate the responses of mammary epithelial cells (Mac-T) challenged with *S. aureus* strains, cell death scoring was performed (Figure 1A, Table 3 and Appendix A). Different multiplicities of infection (MOIs) and infection time points were tested to determine the optimal experimental conditions. Higher MOI levels (4:1 and 10:1) and longer infection durations (24 h) resulted in more severe cell death, with the most pronounced effects observed in the clinical mastitis group. Based on these results, an MOI of 10:1 and a 6 h infection period were selected as the optimal conditions for subsequent transcriptional analysis. This choice provided significant cell death while minimizing bacterial overgrowth (Figure 1A).

The principal component analysis (PCA) showed distinct clustering of Mac-T cells challenged with *S. aureus* strains from different groups. Along the PC1 axis, cells challenged with strains from the low-SCC group (ML), high-SCC group (MH), and clinical mastitis group (MM) showed progressively greater separation from the control group (C) (Figure 1B). This increasing separation reflects the transcriptional variability among the groups, suggesting group-specific differences. The differential gene expression analysis showed a progressive increase in the number of DEGs across the groups (Figure 1C). Strains from the clinical mastitis group induced the most extensive transcriptional changes, with an average of 4555 DEGs, compared to 2835 DEGs in the high-SCC group and 1189 DEGs in the low-SCC group.

A total of 182 DEGs were consistently identified across the ML, MH, and MM groups compared to the control group (Figure 1D, Appendix A). These DEGs may represent potential candidate markers for the host response to *S. aureus* infection. Among these, a cluster of SNPs significantly associated with SCS was identified within the *DCK* gene by examining the GWAS summary data (Figure 1E). Following infection, the expression of the *DCK* gene was markedly upregulated. Moreover, strains isolated from the clinical mastitis group induced a higher expression of *DCK* compared to strains obtained from the other two groups. Within the clinical mastitis group, strains MM1, MM2, and MM3 induced distinct DEG sets, indicating strain-specific transcriptional responses (Figure 1G).

### 2.3. Common DEGs and Key Modules in the WGCNA Are Enriched in the Ferroptosis Pathway

WGCNA identified distinct gene expression modules associated with *S. aureus* challenge from different SCC and mastitis groups (Figure 2A). A comparison between the control group and the clinical mastitis group revealed significant correlation shifts in several modules. For instance, modules 7 and 8 transitioned from weakly negative correlations in the control group to strongly positive correlations in the MM group. In contrast, modules 17 and 18 showed the opposite trend, shifting from weakly positive correlations in the control group to strongly negative correlations in the MM group, indicating substantial regulatory changes associated with infection progression.

Modules significantly associated with the clinical mastitis group (modules 7, 8, 9, 10, 17, and 18) were subjected to KEGG pathway enrichment analysis (Figure 2B). Module 8 showed significant enrichment in pathways related to autophagy and ubiquitin-mediated proteolysis, while module 9 was enriched in pathways related to endocytosis and tight junctions, suggesting their roles in cellular stress responses and intercellular communication. Modules 17 and 18 were significantly enriched in pathways associated with immune responses, inflammatory processes, and cell death mechanisms, indicating their involvement in the molecular changes observed in mastitis.

GWAS enrichment analysis was performed to assess the associations between DEGs and bovine health traits (Figure 2C). The analysis revealed significant enrichment for health-related traits, with notable associations observed for mastitis (MAST). Specifically, in pairwise comparisons (e.g., C vs. ML1, C vs. MH1, and C vs. MM1), multiple DEGs showed significant associations with MAST. These associations were most prominent in the clinical mastitis (MM) and high-SCC (MH) groups, indicating that *S. aureus* strains from these groups elicited host responses linked to mastitis susceptibility.

Additionally, the KEGG enrichment analysis revealed that DEGs identified across the ML, MH, and MM groups or unique to specific groups were significantly enriched in the ferroptosis pathway (Figure 2D, Appendix A). This suggests that ferroptosis is a key pathway involved in the host cellular response to *S. aureus* challenge, regardless of strain-specific differences.

### 2.4. Expression Patterns of Ferroptosis-Related Differentially Expressed Genes (FRGs) in S. aureus Mastitis

The expression patterns of FRGs were analyzed to investigate their relationship with the severity of bovine mastitis induced by *S. aureus*. As shown in Figure 3A, 11 FRGs exhibited consistent expression changes compared to the control group, with more pronounced alterations in the clinical mastitis (MM) group.

Among these genes, *HMOX1*, *SAT1*, and *VDAC2* (Figure 3B) were significantly upregulated across all challenged groups, with the highest expression levels observed in the MM group. In contrast, *SLC11A2* and *STEAP3* (Figure 3D) were consistently downregulated, indicating reduced iron uptake and altered iron homeostasis in the challenged cells. The correlation analysis of FRGs (Figure 3C) revealed strong inter-gene correlations, indicating possible coregulation or functional relationships among these genes. The PPI network (Figure 3D) further confirmed interactions between these FRGs, highlighting their involvement in a coordinated response. Additionally, a comparative analysis between the control and MM groups (Figure 3E) indicated ferroptosis activation during *S. aureus* infection. Genes such as *SLC11A2*, *SAT1*, *HMOX1*, and *VDAC2* were identified as significant drivers of ferroptosis, as confirmed by comparison with the FerrDb database (http://www.zhounan.org/ferrdb, accessed on 10 August 2024). These findings suggest that ferroptosis-related genes play a critical role in the cellular response to *S. aureus* infection, with ferroptosis activation being the most pronounced in the MM group.

### 2.5. Ferroptosis Induced by S. aureus Challenge in Mac-T Cells and Its Inhibition by Ferrostatin-1

Ferroptosis induction in Mac-T cells challenged with *S. aureus* strain M2 (clinical mastitis group) and its inhibition by Ferrostatin-1 (Fer-1) were evaluated. Following the M2 challenge, reactive oxygen species (ROS), a key marker of ferroptosis initiation, were significantly elevated (Figure 4A). Additionally, a reduction in calcein fluorescence indicated an expanded labile iron pool (LIP), confirming ferroptosis activation in M2-challenged Mac-T cells (Figure 4B).

To assess the inhibitory effect of Fer-1, Mac-T cells were treated with Fer-1 at concentrations of 10 µM and 20 µM. Fer-1 treatment significantly reduced ROS levels in a dose-dependent manner, with 20 µM showing the most effective inhibition (Figure 4A). Subsequent LIP measurements demonstrated that 20 µM Fer-1 effectively mitigated iron accumulation, as reflected by restored calcein fluorescence (Figure 4B). Fluorescence microscopy further validated these quantitative findings, showing a notable decrease in ROS and LIP fluorescence intensity after Fer-1 treatment compared to M2-challenged cells, visually confirming its ability to alleviate oxidative stress and iron overload (Figure 4C).

Changes in ferroptosis-related gene expression were also analyzed. *S. aureus* challenge significantly upregulated ferroptosis-related genes, including *VDAC2*, *ACSL4*, *HMOX1*, and *SAT1*, while downregulating *SLC3A2* expression (Figure 4D). Fer-1 treatment notably reduced the expression of upregulated ferroptosis-related genes and restored *SLC3A2* expression, further confirming its inhibitory role in ferroptosis.

To further validate our cellular findings, the RNA-seq analysis of milk somatic cells from cows with *S. aureus*-induced subclinical mastitis (BioProject PRJNA878880) revealed significant upregulation of *SAT1* expression (Figure 4E). Furthermore, the analysis of transcriptomic data from mother–daughter pairs affected by subclinical mastitis demonstrated elevated expression of both *SAT1* and *HMOX1* in the infected groups compared to the healthy controls. This trend was consistently observed across two generations: *S. aureus*-infected mothers (SMs) versus healthy mothers (HMs) and their *S. aureus*-infected daughters (SMDs) versus healthy daughters (HMDs) (Figure 4F). The intergenerational concordance in FRG dysregulation reinforces our in vitro observations, suggesting that ferroptosis activation is a conserved mechanism in *S. aureus* mastitis pathogenesis.

## 3. Discussion

### 3.1. Conserved Host Ferroptosis Response

The progression of *S. aureus*-induced bovine mastitis is closely linked to the dysregulated activation of host ferroptosis, a form of iron-dependent cell death driven by lipid peroxidation. This study combines multi-omics analyses to demonstrate that ferroptosis plays a key role in host–pathogen interactions during *S. aureus* infection. Whole-genome sequencing of 11 *S. aureus* strains identified three dominant sequence types (ST1, ST97, and ST398), which align with their epidemiological prevalence in China [24]. However, no direct correlation was observed between bacterial genotypes (e.g., cap8 virulence genes) and disease severity [25], suggesting that host responses, rather than bacterial virulence factors, dominate disease progression [26,27]. Transcriptomic profiling of *S. aureus*-challenged Mac-T cells showed a progressive increase in DEGs as bacterial strains from low to high SCC and clinical mastitis stimulated the cells, with ferroptosis-related genes (*HMOX1*, *SAT1*) consistently upregulated across all stages. This upregulation was further validated in the mother–daughter pairs, where both the infected dams and calves exhibited higher blood *HMOX1*, *SAT1* levels compared to the healthy controls. These findings align with emerging evidence that ferroptosis regulates iron metabolism and oxidative stress during bacterial infections [28].

### 3.2. Dual Pathways Triggering Ferroptosis

Ferroptosis activation in *S. aureus*-infected mammary tissue involves two interconnected pathways. First, bacterial toxins, such as α-hemolysin, directly impair host cystine uptake by downregulating *SLC7A11*, a key transporter for glutathione synthesis. This depletion of glutathione inactivates GPX4, a critical antioxidant enzyme, leading to uncontrolled lipid peroxidation [29]. While similar mechanisms are seen in *Salmonella* infections (via the PhoP/Q system), the process in *S. aureus* appears more complex, possibly involving quorum-sensing systems like Agr for fine-tuned regulation [30,31]. Second, host iron overload creates a self-amplifying loop: *HMOX1*-mediated heme degradation releases free iron [32], which fuels Fenton reactions to generate ROS [33]. ROS further activates *SAT1*, a polyamine oxidase that oxidizes spermine into H_2_O_2_, exacerbating oxidative stress and lipid peroxidation. This “iron–ROS–ferroptosis” cycle is particularly destructive in lactating mammary glands, where iron-rich milk synthesis provides abundant substrates for peroxidation [34,35]. The convergence of these pathways explains the consistent upregulation of HMOX1 and SAT1 across infection stages.

### 3.3. Interaction with Other Programmed Cell Death Pathways

While apoptosis and pyroptosis are common in bacterial infections, *S. aureus* mastitis exhibits unique dynamics. Pyroptosis, mediated by the NLRP3 inflammasome and gasdermin D (*GSDMD*), was suppressed in infected cells (Appendix A). This suppression likely reflects bacterial immune evasion strategies, as *S. aureus* nucleases degrade extracellular DNA to inhibit inflammasome activation—similar to the mechanism used by Mycobacterium tuberculosis via the ESX-1 system [36,37,38]. In contrast, apoptosis exhibited divergent patterns between mothers and daughters: *CASP3* mRNA was upregulated in the infected daughters but downregulated in the mothers. Chronic exposure in mothers may drive adaptations favoring anti-apoptotic signals to maintain tissue integrity, whereas daughters, still in early stages of lactation, rely on apoptosis to clear damaged cells [39]. Ferroptosis, however, predominates due to its compatibility with the lactating mammary microenvironment. The gland’s high lipid content and iron availability create an ideal milieu for lipid peroxidation, while ferroptosis-derived damage-associated molecular patterns (DAMPs) like *HMOX1* recruit neutrophils via HIF-1 signaling [40]. However, excessive inflammation from uncontrolled ferroptosis can impair tissue repair, highlighting its dual role as both a defense mechanism and a driver of pathology.

### 3.4. Breeding Potential of Ferroptosis-Related Genes

Ferroptosis-related genes *HMOX1* and *SAT1* show stable expression across different infection models and strains, making them promising targets for genetic selection. However, *HMOX1*’s dual role—reducing oxidative stress while exacerbating iron toxicity—requires careful balancing. High *HMOX1* expression is associated with reduced SCC, but it also increases milk iron levels, potentially compromising milk quality. This finding aligns with studies in cardiovascular models, where excessive ROS triggers ferroptosis, influencing disease progression [41]. Thus, breeding strategies should prioritize “moderate-expression” haplotypes to optimize disease resistance without sacrificing production traits [42]. Additionally, non-ferroptosis genes like *DCK*, identified as SCC-linked markers in genetic studies, require further validation to clarify their role. For example, functional validation studies have explored the protective role of miR-223 in *S. aureus*-induced bovine mastitis [43]. Similarly, SNP analysis of the *TLR4* promoter has identified associations between specific polymorphisms and susceptibility to bovine subclinical mastitis, highlighting the potential of genetic selection in improving disease resistance [44]. These studies validate the utility of genetic markers for mastitis resistance and support the application of similar approaches to optimize ferroptosis-related genes in selective breeding strategies.

### 3.5. Therapeutic Limitations and Alternative Strategies

Despite Ferrostatin-1′s efficacy in vitro, its agricultural application faces significant barriers. The compound has low metabolic stability and poor penetration into mammary tissue, limiting its practical use [45], while formulation strategies like liposomal encapsulation remain cost-prohibitive for large-scale farming. Off-target effects, including NF-κB suppression and delayed bacterial clearance, further complicate its utility. As an alternative, genetic selection based on *HMOX1* and *SAT1* offers a more sustainable strategy. Future research will focus on identifying genetic regulatory elements that control *HMOX1* expression, with the aim of validating these findings for use in genetic selection programs to enhance disease resistance in dairy cows.

### 3.6. Limitations and Future Directions

Cell culture models have limitations, as they cannot fully replicate the complexity of the in vivo environment. To address this, we validated our findings using RNA-seq data from milk samples of cows with subclinical *S. aureus* mastitis and healthy cows, along with blood leukocyte transcriptome data from cow–mastitis pairs. These data confirmed the upregulation of the ferroptosis marker gene *SAT1*. However, several limitations remain. First, although RNA-seq data confirmed *SAT1* upregulation in subclinical mastitis, direct evidence of ferroptosis in vivo is still lacking. Second, the role of the mammary microbiome was not explored, leaving gaps in our understanding of host–microbe interactions. Third, the small sample size of only four cow–mastitis pairs in the *S. aureus* infection group may limit the robustness of the findings. Future studies should focus on validating ferroptosis markers in vivo, conducting protein-level analyses in relevant models, and confirming these results at the individual cow level. Finally, increasing the sample size in subsequent trials will be crucial to strengthening the reliability and applicability of the findings.

## 4. Materials and Methods

### 4.1. Experimental Design and Sample Information

To elucidate the molecular mechanisms underlying bovine mastitis induced by *S. aureus*, an in vitro model involving bovine mammary epithelial cells (Mac-T cells) was employed in this study (Appendix A). Eleven *S. aureus* strains were isolated from fresh milk samples obtained from Holstein cows and classified into three groups based on somatic cell count (SCC) and clinical status of cows, including low SCC (L1-L4), high SCC (H1-H4), and clinical mastitis (M1-M3). Mac-T cells were challenged with these strains as follows: the ML group was exposed to low SCC strains (L1-L4), the MH group to high SCC strains (H1-H4), and the MM group to clinical mastitis strains (M1-M3). Control group C was treated with phosphate-buffered saline (PBS). Each experimental group, including the control, consisted of three biological replicates, yielding a total of 36 samples for RNA sequencing analysis.

RNA-seq data generated from the challenged Mac-T cells were analyzed using principal component analysis (PCA), differential gene expression analysis, and weighted gene co-expression network analysis (WGCNA). KEGG and GWAS enrichment analyses were performed on the DEGs and key modules. To validate the findings of the in vitro model, publicly available RNA-seq data from milk somatic cells of cows with subclinical mastitis and healthy controls were used to assess whether the expression trend of ferroptosis-related genes was consistent with observations at the individual cow level. In addition, intracellular levels of ROS and LIP were measured as indicators of ferroptosis in Mac-T cells.

### 4.2. S. aureus Sample Collection

Milk samples were collected from 1112 lactating Holstein cows across seven farms located in northern China. For each cow, 30 mL of fresh milk was aseptically collected and mixed from all four lactating quarters. To minimize contamination risk, the samples were transported to the laboratory within 4 h under refrigeration.

Upon arrival, 10 µL of each milk sample was streaked onto blood agar plates (BAPs) and incubated at 37 °C for 24–48 h to evaluate sample purity and detect any possible contamination. The samples with more than two distinct colony types were considered contaminated and excluded from subsequent analysis. The colonies showing typical characteristics of *S. aureus* were further subcultured for isolation.

Presumptive *S. aureus* colonies were identified based on colony morphology, which included typical hemolytic activity on blood agar plates. These colonies were subsequently subcultured on tryptic soy agar (TSA, HB0177-1, Hopebio, Qingdao, China) plates for purification. The purified colonies were then confirmed as *S. aureus* using Gram staining and coagulase testing. A total of 191 *S. aureus* isolates were confirmed [46], and from these, 11 strains were randomly selected for further analysis in this study.

The confirmed isolates were cultured overnight in tryptic soy broth (TSB), supplemented with 20% sterile glycerol (ST1353, Beyotime, China), and stored in cryovials at −80 °C until further use.

### 4.3. Cell Culture and S. aureus-Challenged Mac-T Cells

Mac-T cells were recovered and cultured for three passages at 37 °C in a humidified environment with 5% CO₂. The cells were maintained in Dulbecco’s Modified Eagle Medium (DMEM) supplemented with GlutaMAX (D5193, ThermoFisher Scientific, Waltham, MA, USA), 10% fetal bovine serum (12484028, Gibco, Grand Island, NY, USA), and 100 U/mL penicillin–streptomycin (15070063, Gibco, Grand Island, NY, USA). The Mac-T cells were seeded into six-well plates (Corning, NY, USA) at a density of 5 × 10⁵ cells per well.

To determine the optimal challenge conditions, the Mac-T cells were exposed to 11 *S. aureus* strains under different conditions, including various time points (6, 12, and 24 h) and multiplicities of infection (MOIs of 1:1, 4:1, and 10:1). Each condition was performed in six replicates. Post-challenge, cell morphology was assessed microscopically, and the severity of cell death was categorized into five grades using a scoring system adapted from previous studies (Table 4).

For the subsequent experiments, once the Mac-T cells reached approximately 80% confluence, the medium was replaced with antibiotic-free DMEM. The cells were then challenged with the 11 *S. aureus* strains at an MOI of 10:1 for 6 h, while the control groups were treated with PBS. All conditions were performed in triplicate. Following the incubation period, the cells were washed three times with sterile PBS to remove non-adherent bacteria. Subsequently, 1 mL of TRIzol reagent (Invitrogen, Carlsbad, CA, USA) was added to each well to preserve and extract total RNA, which was then used for RNA sequencing library construction.

### 4.4. Bacterial Whole-Genome Sequencing (WGS) and Analysis

Bacterial DNA preparation and extraction. Eleven S. aureus strains were retrieved from storage at −80 °C and thawed at room temperature. Then, 50 microliters of bacterial stock was inoculated into 5 mL of TSB and cultured at 37 °C with shaking at 200 rpm for 12 h. Following culture, 50 µL of bacterial solution was serially diluted six times, plated on solid medium, and incubated at 37 °C for 12 h. Single colonies were isolated and cultured in TSB. Genomic DNA was extracted from these cultures using a Wizard^®^ Genomic DNA Purification Kit (Promega, Madison, WI, USA), according to the manufacturer’s protocol.

Genomic DNA sequencing. The integrity and purity of genomic DNA were assessed by 1% agarose gel electrophoresis. DNA concentration was quantified using a Qubit^®^ DNA Assay Kit on a Qubit^®^ 3.0 Fluorometer (Invitrogen, USA). For library construction, 0.2 µg of genomic DNA per sample was utilized. Sequencing libraries were prepared using the NEBNext^®^ Ultra™ DNA Library Prep Kit and subsequently sequenced on the Illumina NovaSeq 6000 platform (Novogene, Beijing, China).

Quality control, sequence assembly, and MLST typing. Raw sequencing reads were subjected to quality control using Trimmomatic version 0.38. The reads containing more than 30% low-quality bases were discarded, and the reads with an average quality score below 30 were trimmed using a sliding window of 3 bp. The quality-filtered reads were then assembled into contigs using SPAdes (version 3.15.2) with multiple k-mer values (55, 65, and 69) to optimize assembly performance. Assembly integrity was assessed using Quast. Multilocus sequence typing (MLST) was conducted based on seven housekeeping genes, including *arcC*, *aroE*, *glPf*, *gmk*, *pta*, *tpi*, and *yqiL*. Allelic profiles and sequence types (STs) were determined using the MLST database (https://pubmlst.org/saureus/, accessed on 10 November 2022).

Virulence factor analysis. The virulence factor was identified using the VirulenceFinder 2.0 database. Specific virulence genes, including the cap8 family genes (*cap8e*, *cap8g*, *cap8l*, *cap8o*, *cap8p*) and the cytolytic pore-forming toxin gene (*α-toxin*), were examined.

### 4.5. RNA Isolation, RNA Sequencing, and RNA-Seq Data Analysis

RNA Isolation, Sequencing, and Quality Control. Total RNA was extracted from the Mac-T cell samples using TRIzol reagent (Invitrogen, USA), according to the manufacturer’s instructions. RNA integrity was assessed by electrophoresis on 1.5% agarose gels, followed by quantification using a NanoDrop 2000 spectrophotometer (ThermoFisher Scientific, USA) and verification using a Qubit 2.0 Fluorometer (Invitrogen, Carlsbad, CA, USA). RNA integrity was further assessed using an Agilent 2100 Bioanalyzer (Agilent Technologies, Santa Clara, CA, USA), and only the samples with an RNA integrity number > 7 were used for downstream library construction. Stranded mRNA sequencing libraries were prepared using an RNA library preparation kit and sequenced on the Illumina NovaSeq 6000 platform (Novogene, Beijing, China). Initial quality assessment of raw sequencing data was conducted using FastQC (version 0.11.8), with the trimming of adapter sequences and low-quality reads carried out using Trimmomatic version 0.38. The clean reads were then subjected to downstream analyses.

Read Alignment, Abundance Estimation, and Differential Expression Analysis. The clean reads were aligned to the bovine reference genome (ARS-UCD1.2) obtained from the Ensembl database using HISAT2 (version 2.1.0). Sorting and indexing were conducted with Samtools version 1.9, and gene expression quantification was performed using FeatureCounts. Differential gene expression analysis was carried out using the DESeq2 package (version 1.28.1) in R. Genes were considered differentially expressed with an absolute fold change > 1.5 and an adjusted *q*-value < 0.05. Candidate marker genes were identified based on consistent differential expression across all 11 *S. aureus* strains.

Weighted gene co-expression network analysis (WGCNA). The WGCNA package (v1ersion 1.73) [47] in R was used to construct a gene co-expression network from the 8000 genes with the highest median absolute deviation (MAD). The co-expression similarity was used to convert the adjacency matrix to a topology overlap matrix (TOM), and the genes were clustered using the 1-TOM metric. Modules were merged using a CutHeight threshold of 0.9. The module most significantly associated with the experimental conditions was identified by analyzing differential expression between the challenged and control groups.

### 4.6. Functional Analysis

KEGG enrichment analysis. KEGG enrichment analysis was conducted to identify the biological functions and pathways associated with the DEGs and gene modules obtained from WGCNA. The analysis was performed using the KEGG Orthology-based Annotation System (KOBAS 3.0.3, http://kobas.cbi.pku.edu.cn/genelist/), with a significance threshold of *p* < 0.05. The top 20 significantly enriched KEGG pathways were visualized using the R packages ImageGP (version 2.0) and ggplot2 (version 3.5.1).

GWAS enrichment analysis. GWAS summary data for 27,143 cattle, covering 44 traits related to production, reproduction, and health, were obtained from Figshare (https://figshare.com/s/ea726fa95a5bac158ac1, accessed on 6 July 2024). Single nucleotide polymorphisms (SNPs) within a 10 kb upstream and downstream region of each candidate gene were screened, and their corresponding association *p*-values were extracted from the GWAS summary data [48].

For the enrichment analysis, a hypergeometric test [49] was employed to assess the overrepresentation of significant SNPs within the candidate gene sets according to Equation (1):(1)$$P\left(X\geq k\right)=\sum_{i=k}^{\min\left(n,K\right)}\frac{\left(\begin{array}{c} K\\ i \end{array}\right)\left(\begin{array}{c} N-K\\ n-i \end{array}\right)}{\left(\begin{array}{c} N\\ n \end{array}\right)}$$
where *n* is the total number of SNPs used in the GWAS analysis of the corresponding trait, *K* is the total number of SNPs potentially associated (with GWAS *p*-values < 0.01) with the corresponding trait, *n* is the number of SNPs within a 10kb flanking region of each candidate gene (candidate SNPs), and *k* is the number of candidate SNPs potentially associated with the corresponding trait.

To control for false positives from multiple analyses between candidate SNPs with multiple traits, a multiple testing correction was applied using the FDR method with the Benjamini–Hochberg procedure. An FDR-adjusted *p*-value of less than 0.05 indicated a significant association between the gene set and the corresponding trait.

### 4.7. Ferroptosis-Related Assays and Analysis

ROS and LIP detection in Mac-T cells with Ferrostatin-1 treatment. The Mac-T cells were challenged with *S. aureus* for 6 h, with or without 20 μM Ferrostatin-1, a ferroptosis inhibitor. After the challenge, the cells were washed twice with PBS and treated with 0.1 µM CA-acetoxymethyl ester (C2012, Beyotime, China) at 37 °C for 30 min to assess the labile iron pool (LIP). The cells were subsequently washed and treated with 100 μM deferiprone for 45 min or left untreated for LIP quantification. LIP levels were determined by measuring fluorescence intensity using a microplate reader (excitation at 488 nm, emission at 525 nm). The difference in fluorescence between deferiprone-treated and untreated samples represented the LIP level [50].

Quantitative real-time PCR (qRT-PCR) for ferroptosis-related differentially expressed genes (FRGs) post-treatment. Total RNA extracted from the treated Mac-T cells, as described in the RNA sequencing section, was used to quantify FRG expression. cDNA was synthesized, and qRT-PCR was performed in triplicate for each sample. The average cycle threshold (Ct) values were used for analysis, and relative gene expression levels were calculated using the 2^−ΔΔCt^ method to assess the impact of Ferrostatin-1 on gene expression following the *S. aureus* challenge. Primers used for qRT-PCR are listed in Appendix A.

Correlation analysis of FRGs. Pearson correlation coefficients between FRGs were calculated using the cor.test function in R. The correlation results were ranked and visualized with hierarchical clustering and the Corrplot package (version 0.92) in R.

Protein–protein interaction (PPI) network construction of FRGs. The STRING database (https://string-db.org/, accessed on 20 August 2024) was used to construct the PPI network among FRGs. PPI data were extracted from the database and filtered to include only interactions among the FRGs under investigation, ensuring the network focused on ferroptosis-associated proteins.

### 4.8. Statistical Analysis

Statistical analyses for bioinformatics data were performed using R (version 4.0.1), with detailed methods for WGS and RNA-seq analyses provided in the corresponding sections. For the differential expression analysis of FRGs and measurements of ROS and LIP levels, *t*-tests were conducted using GraphPad Prism (version 9.3.1). Cell death scores, being non-parametric data, were analyzed with the Kruskal–Wallis H test. A significance level of *p* < 0.05 was applied to all tests. Statistical significance is denoted as follows: ns (not significant), * *p* < 0.05, ** *p* < 0.01, *** *p* < 0.001, and **** *p* < 0.0001.

## 5. Conclusions

This study revealed the complex transcriptional responses to various *S. aureus* strains in bovine mastitis, with a focus on the role of ferroptosis. *HMOX1*, *SLC11A2*, *STEAP3*, *SAT1*, and *VDAC2* were identified as stable molecular markers that are consistent across different *S. aureus* strain challenges. *S. aureus* induces ferroptosis in Mac-T cells, characterized by changes in superoxide anion levels and the labile iron pool. It is mediated by the upregulation of *HMOX1* and *SAT1* and can be effectively reversed by the ferroptosis inhibitor Ferrostatin-1. The SNPs within these ferroptosis-related genes could serve as key genetic markers for molecular breeding strategies aimed at enhancing resistance to mastitis in dairy cattle.

## Figures and Tables

**Figure 1 ijms-26-02506-f001:**
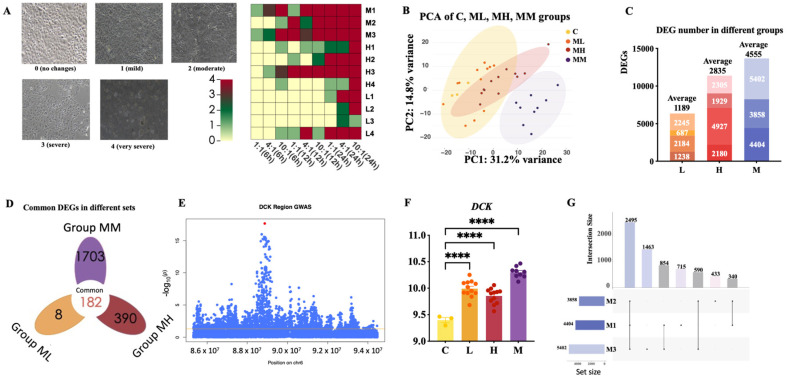
Differentially expressed genes analysis in Mac-T cells challenged with various *S. aureus* strains. (**A**) Scoring schematic diagram of the bacterial infection severity. Infection severity was divided into 0–4 levels. (**B**) PCA results of RNA expression among the M, H, L, and control groups. (**C**) Number of DEGs for four strains in the M, H, and L groups. (**D**) Venn diagram of DEGs across the L, H, and M strain groups. (**E**) Identification of a cluster of SNPs significantly associated with mastitis traits within the *DCK* gene. Blue dots represent SNP loci, red dot represents the most significant loci, and the line represents the *p* = 0.01 threshold. (**F**) Identification of differential expression of the *DCK* gene in various groups. Statistical significance was determined using *t*-tests. Levels of statistical significance are indicated as follows: **** *p* < 0.0001. (**G**) Common and specific DEGs among *S. aureus* strains within the M groups.

**Figure 2 ijms-26-02506-f002:**
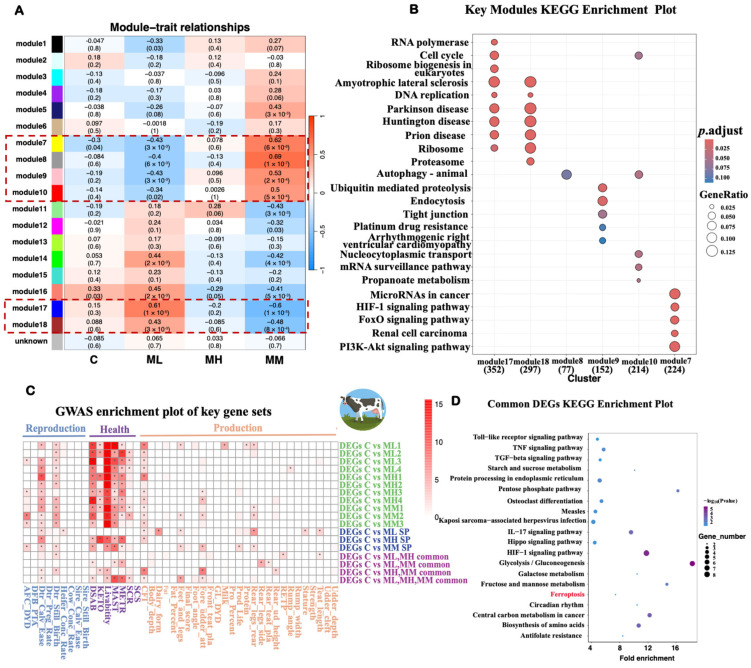
Functional enrichment analysis for the transcriptome data of different experimental groups after *S. aureus* challenge. (**A**) WGCNA analysis. (**B**) KEGG pathway enrichment analysis of genes in the key module. (**C**) GWAS enrichment analysis of key gene sets in bovine large-scale GWAS data; SP: specific DEGs. * indicates *p* < 0.05. The color represents -log(*p*), with more intense red indicating larger values. (**D**) KEGG pathway enrichment analysis of common DEGs; the 20 significant pathways are listed in accordance with the fold enrichment and *p*-value.

**Figure 3 ijms-26-02506-f003:**
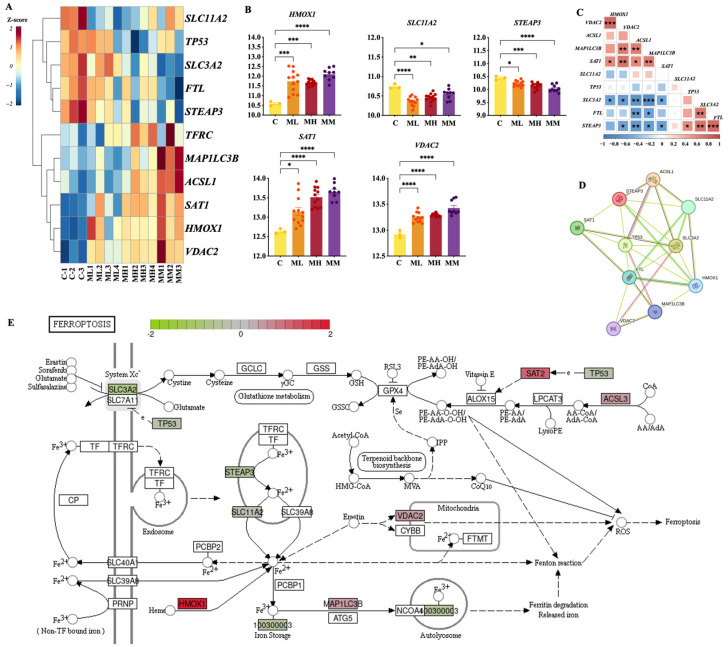
Overall expression features of ferroptosis-related genes challenged by *S. aureus*. (**A**) Heatmap of FRGs in the L, H, and M groups. (**B**) Identification of differential expression of FRGs in various groups of bovine *S. aureus* mastitis. Expression values are shown as VST−normalized counts. Statistical significance was determined using *t*-tests. Levels of statistical significance are indicated as follows: * *p* < 0.05, ** *p* < 0.01, *** *p* < 0.001, and **** *p* < 0.0001. (**C**) Correlation of gene expression among FRGs. (**D**) Protein−protein interaction networks of ten central ferroptosis−related genes in bovines. (**E**) Expression of FRGs in group M on the ferroptosis pathway map. Each box is a gene, and the color presents Log_2_FC between the *S. aureus* challenged group (M group) and the control group.

**Figure 4 ijms-26-02506-f004:**
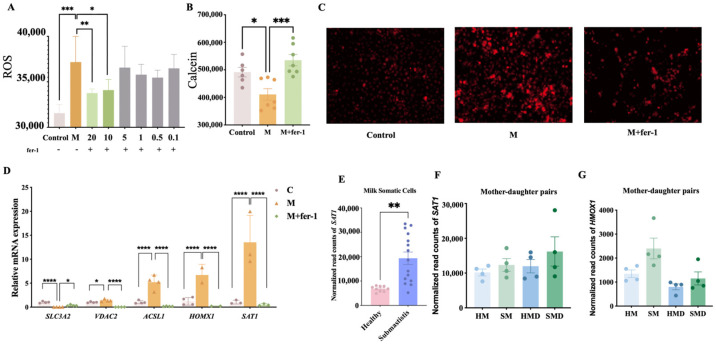
Evaluation of cellular and molecular indicators of ferroptosis in response to *S. aureus* infection and Ferrostatin−1 intervention. (**A**) ROS levels in the control and M2−challenged (M) groups, with Ferrostatin−1 dose effects (0.1−20 μM). (**B**) Scatter plot with bars showing the LIP levels in the control group, the M2 strain−challenged group (M), and the M group treated with Ferrostatin−1 (M + fer−1). (**C**) Fluorescence images depicting cellular states for the control, M, and M + fer−1 groups, demonstrating changes in cellular fluorescence. (**D**) Relative expression of FRGs following treatment with 20μM Ferrostatin−1. (**E**) *SAT1* expression in milk somatic cells from *S. aureus*−infected vs. healthy cows. (**F**,**G**) Expression trends of *SAT1* and *HMOX1* in mother−daughter pairs: *S. aureus*−infected vs. healthy mothers (SM vs. HM) and daughters (SMD vs. HMD). Statistical significance was determined using *t*-tests. Levels of statistical significance are indicated as follows: * *p* < 0.05, ** *p* < 0.01, *** *p* < 0.001, and **** *p* < 0.0001.

**Table 1 ijms-26-02506-t001:** Overview of bacterial WGS data.

Group	Samples	N50	ST	Housekeeping Genes						
				*arcC*	*aroE*	*glpF*	*gmk*	*pta*	*tpi*	*yqiL*
Low SCC	L1	206,226	398	3	35	19	2	20	26	39
Low SCC	L2	244,610	novel	3	1	1	1	1	5	11
Low SCC	L3	6004	novel	-	1	1	2	1	26	-
Low SCC	L4	244,604	97	3	1	1	1	1	5	3
High SCC	H1	15,818	1	1	1	1	1	1	1	1
High SCC	H2	196,295	novel	3	1	1	1	1	5	11
High SCC	H3	92,319	97	3	1	1	1	1	5	3
High SCC	H4	311,676	97	3	1	1	1	1	5	3
Mastitis	M1	204,152	novel	1	-	-	-	1	1	1
Mastitis	M2	317,839	1	1	1	1	1	1	1	1
Mastitis	M3	474,523	1	1	1	1	1	1	1	1

**Table 2 ijms-26-02506-t002:** Presence of virulence genes in *S. aureus* strains.

Gene ^1^	L1	L2	L3	L4	H1	H2	H3	H4	M1	M2	M3
*cap8E*	+ ^2^	+	+	+	+	+	+	+	+	+	+
*cap8G*	+	+	+	+	+	+	+	+	+	+	+
*cap8L*	+	+	+	+	+	+	+	+	+	+	+
*cap8O*	+	+	+	+	+	+	+	+	+	+	+
*cap8P*	+	+	+	+	+	+	+	+	+	+	+
*geh*	+	+	+	+	+	+	+	+	+	+	+
*hly/hla*	+	+	+	+	+	+	+	+	+	+	+
*lip*	+	+	+	+	+	+	+	+	+	+	+
*icaA*	+	+	+	+	+	+	+	+	+	+	+
*aur*	+	+	+	+	+	+	+	+	+	+	+
*hlgA*	+	+	+	+	+	+	+	+		+	+
*map*	+	+	+	+	+	+	+	+		+	+
*hlb*	+	+	+	+	+	+	+	+		+	+
*hld*	+	+	+	+	+	+	+	+		+	+
*adsA*	+	+	+	+	+	+		+	+	+	+
*cap8A*	+	+	+	+	+	+	+	+		+	+
*cap8B*	+	+		+	+	+	+	+	+	+	+
*cap8C*	+	+		+	+	+	+	+	+	+	+
*cap8D*	+	+		+	+	+	+	+	+	+	+
*cap8F*	+	+	+	+		+	+	+	+	+	+
*cap8N*	+	+	+	+	+	+	+	+		+	+
*esxA*	+	+	+	+	+	+	+	+		+	+
*esaA*	+	+		+	+	+	+	+	+	+	+
*esaB*	+	+	+	+	+	+	+	+		+	+
*esaC*		+	+	+	+	+	+	+	+	+	+
*esxB*		+	+	+	+	+	+	+	+	+	+
*isdB*	+	+	+	+	+	+	+	+		+	+
*isdA*	+	+	+	+	+	+	+	+		+	+
*isdC*	+	+	+	+	+	+	+	+		+	+
*isdE*	+	+	+	+	+	+	+	+		+	+
*isdF*	+	+	+	+	+	+	+	+		+	+
*isdG*	+	+	+	+	+	+	+	+		+	+
*icaC*	+	+	+	+	+	+	+	+		+	+
*icaB*	+	+	+	+	+	+	+	+		+	+
*icaD*	+	+	+	+	+	+	+	+		+	+
*icaR*	+	+	+	+	+	+	+	+		+	+
*sspC*	+	+		+	+	+	+	+	+	+	+
*sspA*	+	+	+	+	+	+	+	+		+	+
*ebp*	+	+	+	+	+	+		+	+	+	+
*sbi*	+	+	+	+		+		+	+	+	+
*hlgC*	+	+	+	+	+	+		+		+	+
*cap8M*	+	+		+	+	+	+	+		+	+
*essA*	+	+		+	+	+	+	+		+	+
*hysA*	+	+	+	+	+		+	+		+	+
*lukF-PV*		+	+	+	+	+	+	+		+	+
*srtB*	+	+		+	+	+	+	+		+	+
*fnbA*	+	+	+	+	+	+		+		+	+
*sspB*	+	+	+	+	+	+		+		+	+
*clfA*	+	+	+	+	+	+		+	+	+	+
*hlgB*	+	+		+	+	+		+		+	+
*coa*	+	+	+	+		+		+		+	+
*essB*	+	+	+	+		+		+		+	+
*isdD*	+	+	+	+		+		+		+	+
*fnbB*		+	+	+	+	+		+		+	+
*essC*		+	+	+		+		+		+	+
*spa*	+	+				+		+	+		+
*sdrC*	+	+	+	+		+		+			
*cap8H*			+		+		+			+	+
*cap8I*			+		+		+			+	+
*cap8J*			+		+		+			+	+
*cap8K*			+		+		+			+	+
*sdrD*	+	+	+	+	+	+		+		+	
*clfB*	+	+		+		+		+			
*sdrE*	+	+	+			+	+			+	
*seh*			+		+					+	+
*vWbp*			+							+	+
*inhA*							+		+		
*nheC*							+		+		
*nheB*							+		+		
*nheA*							+		+		
*BAS319*							+		+		
*lukS-PV*							+				
*scn*							+				
*chp*							+				
*sak*							+				
*sea*							+				
Ratio	72.37%	78.95%	72.37%	76.32%	72.37%	77.63%	75.00%	77.63%	36.84%	82.89%	81.58%

^1^ Gene represents the bacterial gene. ^2^ + represents that the strain possesses the gene.

**Table 3 ijms-26-02506-t003:** Nonparametric test results for *S. aureus* severity in Mac-T cells.

	6h Group Medians (P25, P75) ^1^			Kruskal–Wallis *H* Value	*p*
MOI ^2^	L (*n* = 4)	H (*n* = 4)	M (*n* = 3)		
1:1	0.000 (0.0, 1.0)	0.000 (0.0, 0.0)	1.000 (0.0, 1.0)	4.455	0.108
4:1	0.000 (0.0, 1.0)	0.000 (0.0, 0.3)	1.000 (0.0, 2.0)	2.752	0.253
10:1	0.500 (0.0, 3.0)	0.000 (0.0, 1.5)	4.000 (1.0, 4.0)	4.921	0.085

^1^ P25 and P75 are the lower and upper quartiles, respectively. For example, 0.5 (0.0, 3.0) represents the median and upper and lower quartiles. ^2^ MOI presents multiplicity of infection. For example, an MOI of 10:1 indicates that there are ten infections.

**Table 4 ijms-26-02506-t004:** Scoring system for Mac-T cell death severity post-*S. aureus* challenge.

Score	Severity Level	Description
0	No change	Normal cells with no noticeable morphological changes
1	Mild	Slight changes, such as minor shrinkage or swelling
2	Moderate	Chromatin condensation in the nucleus or slight membrane disruptions
3	Severe	Marked nuclear fragmentation and noticeable membrane damage
4	Very severe	Complete structural destruction and significant dissolution of cellular organelles

## Data Availability

The datasets generated and analyzed by the current study are available in the SRA repository (PRJNA913260 and PRJNA913410).

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
