# Peer review of "Ferroptosis-Related Genes as Molecular Markers in Bovine Mammary Epithelial Cells Challenged with Staphylococcus aureus"

_ijms, 2025, doi:10.3390/ijms26062506_

Round 1

Reviewer 1 Report

Comments and Suggestions for Authors

Dear Authors,

Staphylococcus aureus is one of the primary pathogens responsible for bovine mastitis, causing significant economic losses in the dairy industry. This manuscript presents a comprehensive and timely study, particularly in its exploration of ferroptosis as a mechanism of cell death in mastitis. Investigating cell death mechanisms in response to infection holds significant potential for advancing preventive and therapeutic approaches.

The identification of HMOX1, SLC11A2, STEAP3, SAT1, and VDAC2 as potential genetic markers for mastitis resistance is of particular interest. Furthermore, the study highlights the potential of ferroptosis inhibitors, such as Ferrostatin-1, which could offer novel therapeutic strategies. The findings may have implications both for genetic selection in dairy cattle and pharmacological intervention in mastitis treatment.

The hypothesis regarding the role of ferroptosis in bovine mastitis requires further substantiation. The current evidence, based on a single case (L. 49-52), may be insufficient to fully support the claim. A broader discussion on the interplay between ferroptosis and other cell death mechanisms would strengthen the argument.

Greater emphasis should be placed on genetic selection for mastitis resistance in the introduction, particularly considering that selective breeding indices are widely used for this purpose. As the discussion references selection strategies, integrating this aspect earlier in the manuscript would enhance coherence.

The Materials and Methods section should precede the Results, ensuring a logical flow of information. The hypothesis must be clearly linked to the choice of methodology, allowing readers to understand the rationale behind the experimental design.

It remains unclear whether ferroptosis is the primary pathway of cell death in response to S. aureus infection or whether it represents a secondary effect of the inflammatory response. A comparative analysis of ferroptosis alongside apoptosis and pyroptosis in the discussion would provide further clarity.

Overall, the manuscript is scientifically valuable and well-structured, significantly contributing to the understanding of mastitis pathogenesis and genetic resistance markers. The study is noteworthy, particularly if the suggested refinements are incorporated.

Best regards,

Comments on the Quality of English Language

The manuscript is generally well-written and clear. However, some sentences could be refined for better readability and precision, particularly in the discussion section, where complex ideas are presented. A careful language review would help improve the flow and clarity of the text, ensuring a more polished and academically consistent presentation.

Author Response

Staphylococcus aureus is one of the primary pathogens responsible for bovine mastitis, causing significant economic losses in the dairy industry. This manuscript presents a comprehensive and timely study, particularly in its exploration of ferroptosis as a mechanism of cell death in mastitis. Investigating cell death mechanisms in response to infection holds significant potential for advancing preventive and therapeutic approaches.

The identification of HMOX1, SLC11A2, STEAP3, SAT1, and VDAC2 as potential genetic markers for mastitis resistance is of particular interest. Furthermore, the study highlights the potential of ferroptosis inhibitors, such as Ferrostatin-1, which could offer novel therapeutic strategies. The findings may have implications both for genetic selection in dairy cattle and pharmacological intervention in mastitis treatment.

Dear Reviewer,

Thank you for your insightful comments and constructive suggestions, which have significantly improved the quality and clarity of our manuscript. Below, we provide a point-by-point response to your feedback. All revisions in the manuscript are highlighted in red text, and line numbers refer to the revised manuscript with tracked changes.

Comments 1: The hypothesis regarding the role of ferroptosis in bovine mastitis requires further substantiation. The current evidence, based on a single case (L. 49-52), may be insufficient to fully support the claim. A broader discussion on the interplay between ferroptosis and other cell death mechanisms would strengthen the argument.

Response 1: Thank you for your constructive feedback. To address the need for broader substantiation of ferroptosis’ role in mastitis, we have strengthened the hypothesis through three key revisions: 1. Conserved mechanism validation: In the Introduction, we added references to pathogens like Mycobacterium tuberculosis and Salmonella enterica exploiting ferroptosis [10-12], supporting its conserved role in bacterial infections (Lines 42-44). 2. Interplay with other cell death pathways: A new Discussion subsection (3.3) compares ferroptosis with pyroptosis (suppressed via NLRP3/GSDMD inhibition, Fig S2) and apoptosis (divergent CASP3 patterns in mothers/daughters), clarifying why ferroptosis dominates in mammary tissue (Lines 282-298). 3. Multi-level evidence: Public RNA-seq data (BioProject PRJNA878880) and mother-daughter pair analyses (SM/HM; SMD/HMD) confirmed consistent SAT1/HMOX1 upregulation (Figure 4E-F), validating ferroptosis as a stable host response (Lines 227-236).

Comments 2: Greater emphasis should be placed on genetic selection for mastitis resistance in the introduction, particularly considering that selective breeding indices are widely used for this purpose. As the discussion references selection strategies, integrating this aspect earlier in the manuscript would enhance coherence.

Response 2: Thank you for highlighting the importance of genetic selection. In the revised Introduction, we added a dedicated discussion on the role of selective breeding indices and genomic selection (GS) in improving mastitis resistance, emphasizing their integration with functional candidates to optimize breeding strategies (Lines 34-38). In the Discussion, we reinforced this theme by proposing SAT1/HMOX1 as stable genetic markers for selective breeding. A new subsection ("Breeding Potential") links their consistent expression across infection models and mother-daughter pairs to practical strategies (e.g., prioritizing "moderate-expression haplotypes"), balancing disease resistance and milk quality (Lines 300-306). This aligns with earlier examples (e.g., TLR4 SNPs) and ensures coherence between the Introduction’s framework and Discussion’s applications (Lines 311-316).

Comments 3: The Materials and Methods section should precede the Results, ensuring a logical flow of information. The hypothesis must be clearly linked to the choice of methodology, allowing readers to understand the rationale behind the experimental design.

Response 3: Thank you for your suggestion. While the journal’s formatting guidelines prevent relocating the Materials and Methods section, we have revised this section to bring clarity to experimental design.

Comments 4: It remains unclear whether ferroptosis is the primary pathway of cell death in response to S. aureus infection or whether it represents a secondary effect of the inflammatory response. A comparative analysis of ferroptosis alongside apoptosis and pyroptosis in the discussion would provide further clarity.

Response 4: Thank you for raising this important point. To clarify the role of ferroptosis in S. aureus infection, we have added a dedicated subsection ("3.3. Interaction with other programmed cell death pathways") in the Discussion. In this section, we compare ferroptosis with pyroptosis (suppressed by NLRP3/GSDMD inhibition, Fig. S2) and apoptosis (divergent CASP3 patterns in mothers vs. daughters) and show that ferroptosis is important in the lactating mammary gland due to its iron-rich microenvironment. We emphasise that ferroptosis activation is not a secondary inflammatory effect, but a primary pathogenic mechanism driven by bacterial toxin-mediated lipid peroxidation and host iron overload. These findings are consistent with studies demonstrating pathogen-specific modulation of cell death pathways (e.g., Mycobacterium tuberculosis [36-38]). Revisions are highlighted in Section 3.3 (Lines 282-298).

Comments 5: The manuscript is generally well-written and clear. However, some sentences could be refined for better readability and precision, particularly in the discussion section, where complex ideas are presented. A careful language review would help improve the flow and clarity of the text, ensuring a more polished and academically consistent presentation.

Response 5: Thank you for your valuable feedback. To address the concern about readability, we have refined several sentences throughout the manuscript for better clarity and precision. In the Results section, we have made revisions to improve readability (Lines 73-80, 104-105, 109-112, 151-152, 179-180, 195-196, 208-213, and 227-236). In the Discussion section (Lines 250-341), the content has been restructured under six new subheadings to enhance logical flow and ensure critical insights are clearly highlighted: 1. Conserved Host Ferroptosis Response, 2. Dual Pathways Triggering Ferroptosis, 3. Interaction with Other Programmed Cell Death Pathways, 4. Breeding Potential of Ferroptosis-Related Genes, 5. Therapeutic Limitations and Alternative Strategies, 6. Limitations and Future Directions. These revisions aim to improve the flow and clarity of the text, ensuring a more polished and academically consistent presentation.

We sincerely appreciate the reviewer’s thoughtful comments and suggestions, which have greatly contributed to improving the quality and clarity of our manuscript. We have made every effort to address all the concerns raised. However, should there be any further suggestions or adjustments needed, we are happy to make additional changes to ensure the manuscript meets the highest standards for publication in IJMS.

Reviewer 2 Report

Comments and Suggestions for Authors

This manuscript investigates the role of ferroptosis in Staphylococcus aureus-induced mastitis in bovine mammary epithelial cells (Mac-T cells), the article is well structured with results of interest to the reader but some points have to be improved and some experiments have to be added to increase the value of the manuscript

Major points

-Incorporation of invivo study in the dairy cow milk mastitic and healthy cows for ferroptosis marker.

-the role of mastitis is not experimentally confirmed and need more invistigation

-more details on the statistical analysis is needed

-discuss the limitations of Ferrostatin-1 treatment in dairy cows

-results and discussion are dense and need to be more visible to the reader

Some minor points

-the abstract need to be well structured to show the novel findings

-ligand figures are too brief 

-some of the references need to be more updated

-some sentences are too long and complex and need to be separated  

Comments on the Quality of English Language

 Needs minor grammatical refinements, better sentence flow, and conciseness.

Author Response

This manuscript investigates the role of ferroptosis in Staphylococcus aureus-induced mastitis in bovine mammary epithelial cells (Mac-T cells), the article is well structured with results of interest to the reader but some points have to be improved and some experiments have to be added to increase the value of the manuscript

Thank you for your generally positive feedback on the manuscript. We appreciate your recognition of the structure of the article and the interest of the results. We have carefully considered your suggestions and have further improved the experimental design and content to enhance the value of the manuscript. Below, we provide a point-by-point response to your feedback.

Major points

Comments 1: Incorporation of invivo study in the dairy cow milk mastitic and healthy cows for ferroptosis marker.

Response 1: Thank you for highlighting the need for in vivo validation. In the revised manuscript, we have addressed this by incorporating indirect in vivo evidence through two complementary approaches. First, we analysed publicly available transcriptomic data from milk somatic cells of cows with subclinical S. aureus mastitis (BioProject PRJNA878880), which revealed a significant upregulation of the ferroptosis marker SAT1 in infected versus healthy cows (Figure 4E). Second, blood leukocyte transcriptomes from mother-daughter pairs showed intergenerational consistency in ferroptosis-related gene dysregulation (SAT1 and HMOX1 upregulation in SM/SMD vs. HM/HMD groups, Figure 4F) (Lines 227-236). These population-level findings bridge our in vitro results to natural infections. In the discussion, we explicitly acknowledge the limitations of the current study and suggest next steps for future research (Lines 327-340).

Comments 2: the role of mastitis is not experimentally confirmed and need more investigation

Response 2: Thank you for your suggestion. Our study experimentally links ferroptosis to S. aureus-induced mastitis through multiple approaches. The S. aureus strains used were isolated from cows with clinical or subclinical mastitis (high/low SCC), ensuring clinical relevance. In vitro challenge experiments with these strains induced hallmark ferroptosis features in Mac-T cells, including ROS accumulation, labile iron pool (LIP) elevation, and lipid peroxidation (Figure 4A-C). Crucially, Ferrostatin-1 (Fer-1) treatment reversed these effects dose-dependently (Figure 4A-B), directly implicating ferroptosis in cellular damage (Lines 208-226). Data at the individual cow level (Figure 4E-F) further support these results (Lines 227-236). As mentioned in the discussion section, future work will validate these findings in bovine tissues (Lines 337-340). We believe that these experimental results provide strong evidence for the role of ferroptosis in S. aureus-induced mastitis.

Comments 3: more details on the statistical analysis is needed

Response 3: Thank you for your suggestion. We have expanded the Materials and Methods section to provide more details on the statistical approaches used. For the enrichment analysis, a hypergeometric test [41] was used to assess the overrepresentation of significant SNPs within the candidate gene sets (Lines 472-474). Statistical analyses were conducted using t-tests for differential expression analysis of ferroptosis-related genes (FRGs), ROS, and LIP levels. For non-parametric cell death scores, the Kruskal-Wallis H test was used (Lines 510-516).

Comments 4: discuss the limitations of Ferrostatin-1 treatment in dairy cows

Response 4: Thanks. A dedicated subsection in the Discussion now addresses practical barriers to the use of Fer-1 in dairy farming, including poor metabolic stability, limited mammary tissue penetration, and the high cost of advanced formulations (e.g., liposomal encapsulation) (Lines 317-326). We emphasise genetic selection strategies (e.g. SAT1/HMOX1 haplotypes) as a sustainable alternative, in line with the study's focus on breeding applications (Lines 300-307).

Comments 5: The results and discussion are dense and need to be made more visible to the reader.

Response 5: Thank you for your valuable feedback. To address the concern about readability, we have revised the Results section to improve clarity (Lines 73-80, 104-105, 109-112, 151-152, 179-180, 195-196, 208-213, and 227-236). In the Discussion section (Lines 250-341), we have restructured the content under six new subheadings to enhance logical flow and emphasize the visibility of critical insights: 1. Conserved Host Ferroptosis Response, 2. Dual Pathways Triggering Ferroptosis, 3. Interaction with Other Programmed Cell Death Pathways, 4. Breeding Potential of Ferroptosis-Related Genes, 5. Therapeutic Limitations and Alternative Strategies, 6. Limitations and Future Directions.

Some minor points

-the abstract need to be well structured to show the novel findings

Response: Thanks. The abstract has been revised to better highlight the novel findings of the study (Lines 9-13, 15-24).

-ligand figures are too brief 

Response: Thanks. The ligand figures have been expanded for clarity and more detailed presentation (Lines 200, 244-248, and Figure S2-S3).

-some of the references need to be more updated

Response: Thanks. We have restructured the Introduction and Discussion sections for clarity and updated most of the references to include more recent studies.  

 Needs minor grammatical refinements, better sentence flow, and conciseness

Response: Agree. The Abstract now follows a structured format (Objective, Methods, Results, Conclusion) and highlights novel contributions: (1) ferroptosis as a conserved mechanism in S. aureus mastitis and (2) SAT1/HMOX1 as stable genetic markers for breeding.

Reviewer 3 Report

Comments and Suggestions for Authors

In the manuscript "Ferroptosis-related genes as molecular markers in bovine mammary epithelial cells challenged with Staphylococcus aureus”, Xing et al found the correlations with Staphylococcus aureus-induced gene expression and ferroptosis in bovine mammary epithelial cells. Staphylococcus aureus is a major agent and significantly contributes to substantial economic losses in the dairy industry. Thus, finding how and which markers through a broader spectrum of S. aureus strains is essential to potentially reduce this big loss. Through transcriptome analysis, the HMOX1, SLC11A2, STEAP3, SAT1, and VDAC2 were identified as stable molecular markers that are consistent across different S. aureus strains challenge. Overall, this manuscript is worth of publication in ijms.

Many S. aureus-induced ferroptosis are known, and these recent literatures should be mentioned in the Introduction.

Ferroptosis-like cell death and the genes involved should be mentioned in the Introduction.

The cited references in this manuscript are not new and should be followed in mdpi style.

Sentences Lines 67-72, not the Table 1 legend, should be clearly separated. If yes, (Table 1) in the legend should be removed. The housekeeping genes should be explained briefly.

Personally, Figure 5 and Table 4 should be considered to be in Supplemental data. More descriptions should be given.

Other markers involved in ferroptosis-related genes should be compared and discussed.

Author Response

In the manuscript "Ferroptosis-related genes as molecular markers in bovine mammary epithelial cells challenged with Staphylococcus aureus”, Xing et al found the correlations with Staphylococcus aureus-induced gene expression and ferroptosis in bovine mammary epithelial cells. Staphylococcus aureus is a major agent and significantly contributes to substantial economic losses in the dairy industry. Thus, finding how and which markers through a broader spectrum of S. aureus strains is essential to potentially reduce this big loss. Through transcriptome analysis, the HMOX1, SLC11A2, STEAP3, SAT1, and VDAC2 were identified as stable molecular markers that are consistent across different S. aureus strains challenge. Overall, this manuscript is worth of publication in ijms.

Dear reviewer,

Thank you for your positive feedback and constructive suggestions. We have carefully addressed each of your comments to enhance the manuscript’s clarity, relevance, and scientific rigor. Below, we provide a point-by-point response to your feedback.

Comments 1: Many S. aureus-induced ferroptosis are known, and these recent literatures should be mentioned in the Introduction.  

Response 1: Thanks. Added recent studies by Zhao et al. (2023) and Bao et al. (2024), demonstrating that S. aureus may manipulate ferroptosis pathways to promote survival and chronic infection in the mammary gland (Lines 46-48; References 16, 17). In addition, similar to how different viral strains induce strain-specific immune responses in honey bees, our study suggests that different S. aureus strains from cows with different SCC levels may activate conserved host pathways, supporting the role of ferroptosis in mastitis progression (Lines 53-59).

Comments 2: Ferroptosis-like cell death and the genes involved should be mentioned in the Introduction.

Response 2: Thanks. We have now included the concept of ferroptosis-like cell death and the associated genes in the Introduction section, highlighting their relevance to the study of mastitis and their potential role in the pathogenesis of the disease. The revised introduction now includes key regulators of ferroptosis, the role of SAT1 in polyamine metabolism and HMOX1 in iron release, linking their dysregulation to S. aureus-induced inflammation (Lines 48-52; References 18-21).

Comments 3: The cited references in this manuscript are not new and should be followed in mdpi style.

Response 3: Thank you for your suggestion. We have referenced recent research, revised the Introduction and Discussion sections to include more recent research, and cited more recent journals as references (Lines 30-69, 251-343).

Comments 4: Sentences Lines 67-72, not the Table 1 legend, should be clearly separated. If yes, (Table 1) in the legend should be removed. The housekeeping genes should be explained briefly.

Response 4: Thanks for your suggestion. The sentences have been separated from the legend and the order has been adjusted. The changes are reflected in Lines 75-81.

Comments 5: Personally, Figure 5 and Table 4 should be considered to be in Supplemental data. More descriptions should be given. Other markers involved in ferroptosis-related genes should be compared and discussed.

Response 5: Figure 5 (Technical route for this study) and Table 4 (Experimental design for S. aureus strain challenges in Mac-T Cell) have been moved to Supplementary Materials (Figures S3 and Table S3).

Comments 6: Other markers involved in ferroptosis-related genes should be compared and discussed

Response 6: In the Discussion section, specifically in subsection 3.2 ("Dual pathways triggering ferroptosis"), we have elaborated on the roles of key ferroptosis-related genes, including HMOX1 and SAT1, and their involvement in the iron-ROS-ferroptosis cycle. Additionally, in subsection 3.4 ("Breeding potential of ferroptosis-related genes"), we discuss the potential of these genes as targets for genetic selection and compare their expression patterns across different infection models (Lines 284-317). These discussions provide a comprehensive analysis of the markers involved in ferroptosis and their implications in the context of S. aureus infection.

We sincerely appreciate the reviewer’s thoughtful comments and suggestions, which have greatly contributed to improving the quality and clarity of our manuscript. We have made every effort to address all the concerns raised. However, should there be any further suggestions or adjustments needed, we are happy to make additional changes to ensure the manuscript meets the highest standards for publication in IJMS.

Round 2

Reviewer 2 Report

Comments and Suggestions for Authors

The revised manuscript was greatly improved